# "It's a last straw situation"; Overdose response and preferences for seeking support from emergency services during overdose in rural Ohio

David C. Colston[1,2,3], Vivian F. Go[1], Clare Barrington[1,2], Adams L. Sibley[1], Hannah M. Piscalko[4], Laura Limarzi-Klyn[1], William C. Miller[5]

1 Department of Health Behavior, Gillings School of Global Public Health, University of North Carolina – Chapel Hill, Chapel Hill, North Carolina, United States of America, 2 Carolina Population Center, University of North Carolina – Chapel Hill, Chapel Hill, North Carolina, United States of America, 3 Injury Prevention Research Center, University of North Carolina – Chapel Hill, Chapel Hill, North Carolina, United States of America, 4 Division of Epidemiology, College of Public Health, The Ohio State University, Columbus, Ohio, United States of America, 5 Department of Epidemiology, Gillings School of Global Public Health, University of North Carolina – Chapel Hill, Chapel Hill, North Carolina, United States of America

## Abstract

### Objectives

We assessed how people who use drugs in rural Ohio communities affected by the opioid epidemic respond to overdose, their preferences for seeking support from first responders, and factors contributing to their willingness or hesitancy to call 9-1-1 during an overdose.

### Methodology

Semi-structured interviews were conducted in select rural Ohio counties from 2021–2022 as part of the Ohio Opioid Project, one of eight sites in the Rural Opioid Initiative focused on overdose in the rural United States. Forty-four interviews with people who use drugs were coded using thematic analysis.

### Results

Most participants had suffered an overdose (23/43) or witnessed (33/43) an overdose. Most participants preferred not to seek first responder support during an overdose if the victim was able to be revived. As a result, first responder support was not sought in most overdoses experienced or witnessed by participants. When first responders were called, outcomes varied: some participants felt the overdose victim's lives were saved, others felt stigmatized by first responders, and one was arrested by law enforcement at the scene. Among participants who preferred not to seek support from first responders, fear of arrest, hospitalization, and stigmatization were the main reasons.

**Data availability statement:** Data contain potentially identifiable information that could result in loss of anonymity. The especially sensitive nature of the subject matter (illicit substance use) and discussion of experiences with organizations in the community (first responders, law enforcement, judicial system) could leave participants open to legal action or further stigmatization if identified. Taken together, public access of de-identified transcripts would violate protections issued by Ohio State University's Institutional Review Board, and researchers would have to receive permission from the relevant ethics board of approval before viewing entire transcripts. Outside of authors on the study, approval to access data can be sought from Ohio State's Huron IRB System via email (IRBInfo@osu.edu).

**Funding:** This work was supported by National Institutes of Health/National Institute on Drug Abuse Grant Numbers:UG3DA044822/ UH3DA044822/ UH3DA044822-04S1 (Grants awarded to WCM, VFG) https://www.drugabuse.gov/. DCC also received support from the National Center for Injury Prevention and Control (Grant ID: R49CE003568) and National Institute for child Health and Human Development (Grant ID: T32HD091058).

**Competing interests:** The authors have declared that no competing interests exist.

## Significance/Contribution

This study details intervenable factors that could prevent overdoses from escalating to fatalities in rural Ohio, including: increasing awareness and knowledge for Good Samaritan Laws to encourage calls to 9-1-1 during overdose, ensuring unlawful arrests are not made at the scene of an overdose, and de-stigmatizing substance use among first responders and health care professionals. Findings may not be generalizable to the entire United States, but may have applicability in other rural, Appalachian areas.

## Introduction

Drug overdose mortality has peaked in the United States in recent years, topping 100,000 deaths in 2021, 2022, and 2023 [1]. With the concurrent rise in the use of potent synthetic opioids such as fentanyl [2], stimulants [2], and drug adulteration [3–5], the overdose epidemic is likely to remain a considerable public health problem for the foreseeable future. In addition to administering naloxone to an overdose victim if available, the Substance Abuse and Mental Health Service Administration (SAMHSA) recommends calling 9-1-1 to receive medical attention even if those on the scene successfully revive the victim [6], as victims are at risk for re-overdosing up to 90 minutes following naloxone administration.

Many states, including Ohio, have passed Good Samaritan Laws (GSLs) [7], policies that grant some form of protections to people seeking emergency support for an overdose victim. In Ohio, immunity is provided for minor drug possession, but the person seeking immunity may be required to receive a referral to a properly credentialed addiction treatment facility within thirty days and submit proof of doing so [8]. Moreover, immunity can only be used twice, and people under parole or probation are not eligible.

Correct knowledge of GSLs is associated with increased emergency calls for overdose, but the simple presence of GSLs might not be sufficient to drive calls to 9-1-1 [9]. While a GSL is 'on the books' it does not guarantee people are aware of it: one year after Maryland passed its GSL, 81% of Baltimore residents were unaware of its existence [10]. This GSL knowledge gap is not limited to civilians or people who use drugs (PWUD). In 2017, only 26% of law enforcement officials across 20 states were able to identify whether or not the GSL offered citizens protection from arrest at the scene [11]. Still, awareness for GSLs may not be the only issue affecting a PWUD's decision to seek support from first responders during an overdose [12], and a better understanding of decision-making regarding seeking support from first responders during an overdose and related barriers is needed to potentially combat overdose-related deaths.

Importantly, overdose related mortality is not evenly distributed throughout the United States, and deciding whether to call 9-1-1 during an overdose may also vary by region. For example, overdose response preferences are poorly understood in rural Appalachia, an area in which overdose-related mortality in 2021 for adults aged

25–54 was 72% higher than the rest of the country [13]. Appalachia, which holds many counties designated as "High-Intensity Drug Trafficking Areas," [14] is also of particular importance given its historically high prevalence of prescription opioid use [15–17] and the number of overdose deaths attributable to polydrug use [16]. Rural Appalachia also boasts a rich culture that values self-sufficiency and its residents are reluctant to trust authorities or outsiders while simultaneously providing tangible social support for the Appalachian in-group, such as neighbors and family members [16,18]. With the hesitancy to trust authority figures, when faced with an overdose, people in Appalachia may seek support from law enforcement or paramedics less commonly. To reduce drug-related morbidity and mortality in Appalachia, further research is needed to understand how PWUD interact with first responders during an overdose and what motivates them to do so. Ohio, home to 32 Appalachian counties [19], offers an ideal study setting to explore these decision making-processes given its historical experience with pill mills (clinics that overly prescribed opioids in early stages of the overdose epidemic) [16]. Moreover, although overdose mortality rates in Ohio declined significantly from 2024 to 2025 mirroring national trends [20], they remain alarmingly high [20,21].

Here, we describe responses to overdose, experiences seeking support from first responders during an overdose, as well as barriers and facilitators of seeking support from first responders during overdose among a sample of PWUD in rural Appalachian Ohio.

## Materials and methods

### Source study

Data from this study were collected as part of the Rural Opioid Initiative (ROI), an eight-site study (Illinois, Kentucky, New England, North Carolina, Ohio, Oregon, West Virginia, and Wisconsin) to characterize and address the opioid epidemic in rural areas of the United States [22]. Qualitative ROI studies have increased our understanding of an array of subjects related to rural substance use, detailing motivations behind the use of opioids [23] or combinations of specific drugs [24], harm reduction strategies, [25,26] perceptions of medications for opioid use disorder [27,28], and stigma against and between PWUD [29–31].

### Study sample

Our study includes interviews with participants from counties in the Appalachian region of southern Ohio. The study received ethical approval from The Ohio State University Institutional Review Board. To be eligible for the study, people had to be at least 18 years old, reside in one of the counties in which the study was taking place, and have a history of opioid use not as prescribed by a doctor or any injection drug use. Participants were recruited through local health care and harm reduction organizations and through participant referral, in which participants were asked to nominate others that they felt would be interested and meet the inclusion criteria for the study. Study participants received $25 gift cards as a reimbursement for their time. Written informed consent was initially sought from study participants. However, due to restrictions and safety precautions taken during the COVID-19 pandemic, much of the data were collected virtually. In such instances, consent forms were administered orally, and participants provided their verbal informed consent over the phone or video call. The Ohio State IRB was notified and approved of changes in the method of data collection.

### Data collection and preparation

We conducted the first of 46 semi-structured interviews on February 22, 2021, and the last interview on February 17, 2022. We discarded interviews due to poor audio quality (n = 2) and a duplicate interview that occurred due to having multiple interviewers conducting interviews at different times (n = 1). We discovered the repeat-participant when reviewing transcripts, as the participant recounted the same story and individuals in the story by name. Semi-structured interviews included questions about perceived changes related to drug use in the area (e.g., types of drugs used, people using drugs, economics), personal drug use history, and experiences with personally experienced and witnessed overdoses.

We analyzed the data iteratively, reviewing transcripts throughout data collection and incorporating probes on emergent themes, including questions on overdose response and preferences for seeking support from first responders (e.g., calling 9-1-1, taking to hospital) during an overdose. Thus, the data on overdose experiences were not collected in some initial interviews. After data collection, the interviews were transcribed and imported into Dedoose for analysis.

## Qualitative analysis

We identified the topic inductively through iterative analysis [32], and developed a codebook to better understand experiences with overdose, preferences toward overdose response, and experiences with/attitudes toward first responders. We then conducted a thematic analysis, a flexible strategy used to connect similar experiences across participants [33]. All coding was conducted by one researcher (DCC).

We reviewed transcripts and generated initial codes based on the data [34,35]. Coded passages were labeled by theme and exported into a matrix. Themes and related excerpts were shared with members of the research team (CB, VFG) at various stages of analysis. Feedback from CB and VFG was incorporated into DCC's analytic approach, framing, and interpretation of the data. Matrices included quotes by participant to allow for comparison of personal overdose experiences, witnessed overdoses, seeking support from first responders or healthcare providers during personal or witnessed overdose, preferences for calling 9-1-1 during an overdose, rationale for preference, and general attitudes or experiences with first responders or law enforcement. Experiences with overdose and first responders were coded descriptively, in which meaning was derived directly from the text. Transcripts were coded as "Yes" or "No" based on whether a participant reported having experienced or witnessed an overdose – defined during interviews as an "experience with overdosing, which includes if you passed out, turned blue, or stopped breathing from using drugs." Support from first responders or medical care professionals was sought in multiple ways, the most common was calling 9-1-1, though some drove the person who overdosed to the hospital.

Organizing matrices by theme and participant allowed us to analyze passages by comparing like and contrasting themes between participants and in the context of the participant's entire interview. We took this strategy because, with the exception of those firmly closed to seeking support from first responders during an overdose, participants rarely stated a clear preference for or against, and those that did express a preference often qualified their statement or provided accounts or perceptions that ran contrary to their initially stated preference. As such, coding participant preferences for or against seeking first responder support during an overdose was more interpretive, and based on their interview as a whole rather than one isolated quote. We also were interested in factors that may have preceded and contributed to their preferences for/against seeking support from first responders.

## Results

### Overview of experiences with overdose and preferences for overdose response

Overall, 53% (23/43) of participants had experienced an overdose personally; among those, less than half (39%, 9/23) stated that support was sought from first responders on their behalf at least once (Table 1). Additionally, three-fourths (77%, 33/43) of participants reported having witnessed an overdose, 14 (42%) of whom either directly sought first responder support for an overdose victim on at least one occasion, were at the scene of an overdose where first responders arrived, or thought first responder support was sought for the victim.

### Seeking support from first responders as a "last straw" during overdose

Most participants preferred to seek support from first responders only as a "a last straw situation," that is, if lay-responders could not revive the victims on their own. People who seemed to espouse a stronger preference for seeking support from first responders in overdose situations, "If I need to call 9-1-1… I always call em", still qualified their stance as conditional on a true need: "Sometimes you don't even need an ambulance especially, that person is going to refuse treatment… they

**Table 1. Descriptive, interpretive codes on overdose experiences/response in sample of PWUD in Appalachia, Ohio, collected from the Rural Opioid Initiative, 2021-2022.**

| Variable | N | (%) |
|---|---|---|
| **Personal experience of overdose** | (n = 43) | |
| Yes | 23 | (53%) |
| No or Not Stated | 21 | (47%) |
| **Sought Support Personal OD[a,b]** | (n = 23) | |
| Yes | 9 | (39%) |
| No | 14 | (61%) |
| **Witnessed Overdose** | (n = 43) | |
| Yes | 33 | (77%) |
| No or Not Stated | 11 | (23%) |
| **Sought Support Witnessed OD[a,c]** | (n = 33) | |
| Yes | 14 | (42%) |
| No | 19 | (58%) |
| **Preferences for Seeking Support During OD[a]** | (n = 19) | |
| Prefer Yes | 2 | (11%) |
| Mixed Opinions | 2 | (11%) |
| Prefer No | 13 | (68%) |
| Strong No | 2 | (11%) |

[a]Cells do not sum to the total study sample of n = 43 given this is a qualitative research study, and participants were not uniformly asked questions about overdose and response, and among those asked, not all provided an explicit or interpretable answer.

[b]Only among participants who reported having experienced a personal overdose.

[c]Only among participants who reported having witnessed an overdose.

wake back up, then…there's no need to call ambulance." Similarly, several participants described incidents where 9-1-1 was called initially, but after the victim was revived, the 9-1-1 call was canceled. Finally, calling 9-1-1 during an overdose was for two participants was "not something any of us would've considered".

### Preferences for attending hospital post overdose

Many people who experienced or witnessed an overdose tried to minimize contact with medical professionals, especially if the overdose victim was successfully revived: [I] "walked right out of the hospital as soon as they got me up there." Some only visited the hospital when threatened with arrest, or declined to go even after they received encouragement and a ride from their friends.

### Perceptions of law enforcement in the study sample

Participants detailed encounters with and perceptions of local law enforcement that varied widely. Some cited their respect for local law enforcement, detailing positive experiences in which they were met with empathy or received help with their recovery, while others were hesitant initially but had come to appreciate what they do: "I thought they were bullies but… they treated me like I should know I was better than that, that my kids deserve better and don't get me wrong, I don't go and seek a cop-out and say 'Hey, best friend!', no, but I 100% respect what they go through every day and I 100% agree with everything they put me through because I deserved a whole lot worse."

At the same time, several participants held negative sentiment toward police and the local judicial system they viewed as a corrupt "good ole boy's system":

"small town judicial systems fucking suck,"

"We have no chance, it don't matter if you're good or bad around here, the cops are bad."

Participants described being treated "like a piece of shit" by law enforcement, recounting stories where they felt police were excessively aggressive, and sharing first hand experiences in which they believed the judicial system mismanaged their case, "This cop system, this judicial… system has done nothing for me.".

The corruption and negative sentiment toward local law enforcement was enough to inhibit at least one participant's desire to call 9-1-1 in general when help was needed,

"Fuck no… you don't call the police around here", adding, "they're, uh, glass floor people… They don't… believe in women… rights at any cost… It's so backwoods around here; you can't even bring a black person around without someone threatening to shoot him…I don't know where you're from, but this is a small town."

Finally, while not a common theme across participants, one person felt strongly that they did not need to call 9-1-1 in (non-overdose specific) situations where help might be required due to personal liberties and self-sufficiency, "why call them…Some things you know, we are legally allowed to take care of ourselves."

### Encounters with first responders during/post overdose

Several participants felt that first responders' actions saved the life of the overdose victim: "It actually took five things of Narcan from the ambulance to bring me back… I was gone…I was really lucky to be saved that day". Others detailed more negative experiences with first responders "… They treat addicts as if they're nothin'… one was being nice, and he [the other paramedic] was being a dick just for no reason". Judgment toward PWUD from first responders was posited to be the result of compassion fatigue with overdose victims, "they're dealing with it more.. and they're just kind of getting tired of it… they kind of just got to stop taking it personally.". Being met with judgment, however, has an impact on whether or not PWUD want to seek support from first responders, "if you want people to be comfortable with calling… in those instances then like, you should seriously rethink…as a police officer… approaching me like that." Outside of stigma felt when interacting with first responders and the risk of arrest, one participant was shocked that law enforcement were ill-prepared to save someone suffering an overdose, "I had a guy die right in front of me… and the police standing there… I said, 'You can't give him some Narcan? Something to save his life.' He said, 'Nope. We don't carry it.'""

### Pragmatism preeminent in whether to seek first repsonder support during overdose

The most fundamental factor influencing the decision to seek support from first responders during overdose seemed to be the perception of need, in part due to the time-sensitive nature of reviving the overdose victim if they were still unconscious. Participants were often able to revive the victim on their own using CPR, naloxone, or other opioid overdose reversal techniques. They pragmatically reasoned that because the victim was able to be revived, seeking support from first responders was unnecessary:

"She came to herself, I reckon"

"She was alright, then."

Participants felt that PWUD were adequately trained to respond to an overdose, making the need for first responders moot: "we try not to involve them…I mean drug addicts… we're…trained…as well as EMT on reviving people." The results spoke for themselves, and confidence in the ability to reverse an overdose was reinforced by past successes reviving overdose victims, "no, I didn't call 9-1-1…because I've saved many people in the past." Similarly, decisions to seek

support from first responders was contingent on their naloxone supply/usage. One participant that had naloxone elaborated that they wouldn't want to divert government resources that could have been helping others in need. Others called 9-1-1 during an overdose because they did not have naloxone – or their preferred formulation of naloxone – on hand.

## Other facilitators to seeking support from first responders during overdose

**Personal connection.** In addition to not having the resources, training, or ability to revive an overdose victim oneself, eight participants mentioned seeking first responder support during an overdose of someone that has a personal connection to them. One participant who had outstanding warrants in a neighboring county, was arrested shortly after calling 9-1-1 on behalf of their boyfriend, who was suffering an overdose. They felt it was "kind of shitty I went to jail for-for trying to save my boyfriend," but "wouldn't change it for anything," indicating context, coupled with the positive outcome of reviving their boyfriend superseded the negative repercussions they faced in how they would have approached the situation if they knew what was to happen. Other participants expressed their empathy for the overdose victim, and the guilt they would feel if they did not call 9-1-1 and the victim were to pass away was enough incentive for them to call: "I don't want that on me or on my heart."

## Other barriers to calling seeking support from first responders during overdose

**Location.** Location was one barrier to seeking support during an overdose: whether at a trap house (locations drugs are sold) or shooting gallery (shared space where drugs are used) [36,37] where they did not want police attention; at their dealer's house, who pressured them to get them off their property; or on their personal property, "I said let's get the hell out of here, I don't want this in my car."

**Legal status of victim or witness.** Another contextual factor that affected the decision to seek support from first responders was the legal standing of the victim or bystander witnessing the overdose. One participant elected not to call 9-1-1 during overdose as the victim was facing possible legal trouble, while another participant who also was in legal trouble still called 9-1-1 for an overdose victim but hid in the bathroom when first responders arrived out of fear of getting in trouble.

**Fear of facing stigma.** Participants' fear of facing stigma was another barrier to seeking support from first responders or hospital workers:

"they look down to you so bad… They make you feel like you're scum. Yes, we know we're addicts, yes, we know we have a problem, but you can't down us anymore than what we down ourselves…--and they act like we can't hear them in the hallway talking about us when we can hear every word… Like, 'I can't believe they're using drugs. Why would they sink down that far in their life?'… if I want, like, if I want to look down, I'll look down upon myself.".

**Fear of facing legal consequences.** Fear of arrest was a major barrier to seeking support from first responders during an overdose: "I went back to my house not knowing if he lived or died, which I know, he's alive now [I: Yeah.] but I just left, you know, a human being to die because I didn't want to get in trouble". Fear of getting in trouble, even as a bystander, affected general preferences for seeking support as well, "I don't think it would be good if somebody overdosed… if you know, they die, and they're going to try to blame it on me or you know, they'll just say that I said I did it because they do stuff like that." One participant's fear of getting in trouble was so strong they preferred the possibility of dying to calling 9-1-1 if they could not be revived by their spouse,

"she knows better not to call 9-1-1…I mean, you can't call 9-1-1, she gotta save me, she gotta get me back or she gotta let me die, I mean, I don't want to see her get in trouble. So, like, she's got to do what she's got to do, you know what I mean, and she knows that."

Fear of getting in trouble was not limited to immediate arrest, though, as some feared seeking support from first responders might put a target on them longer term, opening them to potential stigma, arrests, or involvement with child protective services (CPS):

> "Participant: I mean, unless we're not coming back, we're not calling. I mean, that's just how- it's just not, it's not worth the risk associated with your name being put on that list.

> "Interviewer: What- What do you mean by that, like the list?

> Participant: Well, like I said, it's just not worth it. You call 9-1-1, and your address gets docked as a drug house, whether it- whether it is or not, you know, they automatically start watching, they start suspecting, they, we have children. So, you know, incidentally, CPS is involved; it doesn't matter. If so, for a lot of people, it's scary."

Fear of legal consequences was expressed despite the Good Samaritan Law, which some participants knew about, "they got the, uh, they got a new law that's like, um, a good citizen law or something [I: Yeah.] where um, you know, you can call, anyone right now [I: Yeah.] [coughs] I mean [if you used to?] everyone would go to jail, but like I said, they got this new change in laws, so you won't go to jail." Still, despite this knowledge, many participants noted trying to do what they could before calling 9-1-1 for help because the police "like to take people to jail" – a direct contradiction of what the participant endorsed the law to be. Another participant was unsure if they would get in trouble for calling 9-1-1 during an overdose, but the lasting effect of PWUDs facing criminal possession charges when calling for support during an overdose was enough to make them hesitant to call today.

## Discussion

We found in rural, Appalachian Ohio, PWUD were largely pragmatic, or "practical as opposed to idealistic" [38], in their decisions to seek support from first responders during overdose, consistent with some [39,40] but not all previously published studies [41]. Pragmatism was rooted in the confidence of their overdose reversal abilities, with many only seeking help as a "last straw" – unable to revive the victim on their own. As a result, multiple participants who had suffered or witnessed an overdose recalled that, at least once, support from first responders was not sought for the overdose victim, similar to what has been seen in other studies [40,42,43].

### Practical barriers to seeking support

While idealistic, involving first responders during an overdose is not always seen as the most practical option. As we and others have observed, PWUD often feel capable reversing an overdose on their own and fear consequences of calling 9-1-1 [12,44,45]. Common consequences include facing judgment from first responders experiencing compassion fatigue [46], being exposed to potential legal repercussions [47], or alerting law enforcement or social services that someone who uses or sells drugs lives at a house [40]. Fear of law enforcement are common [12,44,48], and influenced in part by negative experiences PWUD have had when attempting to get help.

Fear of arrest appears to be minimally alleviated by Ohio's Good Samaritan Law (GSL), which offers protections for those seeking help during an overdose. PWUD are commonly unaware of the GSL or its provisions, though findings differ across studies [10,11,49–51]. According to the Social Cognitive Theory, awareness and accurate understanding of the protections offered by GSL are prerequisites for making informed decisions [52,53]. The best approach to raise awareness of the GSL among PWUD is unclear, but the demonstrated connectivity and information-sharing around risk-reduction practices within PWUD networks [54] may offer a promising approach. "Champions" may also be a promising approach, as it has been effective in implementing health services [55]. PWUD have played an important role in distributing peer support services for people in recovery [56,57] and sharing resources, such as naloxone, to reduce the risk of injury or

death from drug use [45,54,58]. Consequently, they are uniquely positioned to increase awareness for GSLs, especially provided with adequate training to ensure accurate knowledge [57,59].

Still, many PWUD still hesitate to call for help even when they know about the GSL [12,60]. Several solutions to this issue have been suggested. Some people have called for law enforcement not to attend 9-1-1 calls for drug overdoses [60], which may reduce fear of legal implications and less hesitancy to call 9-1-1. Others are wary of this approach— over 75% of a sample of law enforcement officers in Seattle, Washington felt their presence was essential to protect medical personnel [50]. A third of these officers believed their presence was needed to enforce the law—a belief contrary to harm reduction [50]. Yet, some law enforcement officers are open to carrying and administering naloxone [61]. Furthermore, police-based naloxone administration training can be well received, which may result in understanding of opioid overdose, reversal, and naloxone use [62,63].

In addition to these possible avenues to improve overdose response, we recommend the state of Ohio (and states with similar policies) amend their GSL to more adequately protect PWUD. The current GSL, for example, caps the number of times a person can call 9-1-1 for an overdose at two, which is insufficient—many participants expressed that they had revived (or attempted to revive) overdose victims on multiple occasions. Similarly, GSL's should provide immunity to people on probation, as many justice-involved people use drugs [64] and people on probation have a high overdose rate [65]. Coupled with exceedingly high rates of overdose seen among formerly incarcerated people [66,67], attempts to arrest rather than rehabilitate PWUD will adversely affect the current overdose epidemic.

### Personal connection and decisions to seek support

For some PWUD, seeking support from a first responder was seen as the most practical option due to their emotional connection to the overdose victim, even if it meant putting themselves at risk. These PWUD felt seeking support from first responders was necessary when they were unable to revive the overdosed party on their own. In contrast, the perceived risk affiliated with calling 9-1-1 to both the overdose victim and the witness led one couple to form a pact to not call 9-1-1 no matter what – "she gotta let me die". Dyadic fear – for themselves and the overdose victim – contributes to hesitancy regarding calling for help from a first responder [68]. In these scenarios, a decision not to call for help equates to an overdose becoming fatal. PWUD and people with substance use disorders have a high acceptance of death and high rates of passive suicidal ideation [69–71]. The ethics of bodily autonomy for unconscious parties with the express wish for no drastic action (beyond trying to save them on their own) are outside the scope of this paper. That said, these instances show how some PWUD have an intense fear of outsider intervention. These examples also show the highly individual decision-making processes related to decisions to call first responders during an overdose.

### Regional relevance and generalizability

The pragmatic decision made by many to only seek support for overdose victims as a "last straw" speaks to the broader belief that these PWUD from rural Ohio can take care of themselves and each other – a tenet central to the cultural ideology of Appalachia that values both self-reliance and collectivism among kin [18]. In this way, findings from our study – while likely not fully generalizable to the entire United States – may be particularly relevant to PWUD in other Appalachian towns. Moreover, takeaways for improving behavioral health may have broader applicability beyond this region, as many PWUD go out of their way to care for one another, often serving as the first line of defense in overdose response and revival [45]. The success PWUD have had in reviving people who have overdosed, the perceived risks experts ask them to take on, and the context-specific decision-making required in each case suggest we should critically reconsider the blanket recommendation to seek support from first responders during an overdose. While ideal in theory, it is not always practical in reality. This consideration underscores the importance of free and widely accessible naloxone for PWUD and community members, enabling them to effectively intervene when an overdose is witnessed [72].

## Limitations

This study has several limitations. First, we did not collect demographic data on race and ethnicity or involvement in the legal system among interviewed participants. We therefore cannot address the role of race/ethnicity on the role of fear for arrest, the risk of being detained due to warrants in neighboring counties, and experienced racism in the judicial system. We strongly urge more intersectional research [73] to better understand how holding multiple marginalized identities – including being a person who uses drugs – shapes decisions to seek help from first responders during an overdose or or to engage in stigmatized overdose prevention efforts.

Because our study was iterative, data on overdose experiences, decisions to seek support from first responders, encounters with first responders, and reasons for preferences toward calling 9-1-1 were not uniformly asked across respondents. Our iterative approach may have limited the information collected from some participants based on the timing of their interviews.

## Conclusion

In rural Appalachian Ohio, we observed that seeking support from first responders during an overdose was a "last straw situation", although some barriers to seeking support are reversible. Increasing awareness of the Good Samaritan Law is essential for PWUD, their families, and law enforcement. Given that a single negative encounter can shape a person's perception of the benefit of calling 9-1-1, law enforcement and EMS professionals must receive trauma-informed training on approaching PWUD and reversing opioid overdoses. Recognition must also be given to the countless lives saved by PWUD, who often reverse overdoses without assistance from first responders. Given their success in doing so, and the fear of stigma or arrest that may result from a 9-1-1 call, we strongly advocate for the continued accessibility of affordable naloxone to laypersons (non-first responders) and PWUD. Finally, language in Good Samaritan Laws nationwide should address the needs of the entire PWUD community, including those requiring multiple overdose reversals and those on parole, to ensure their safety and wellbeing.

## Supporting information

**S1 Data. Sample codebook last straw.**
(XLSX)

## Author contributions

**Conceptualization:** David C. Colston, Clare Barrington.

**Data curation:** David C. Colston, Adams L. Sibley, Hannah M. Piscalko.

**Formal analysis:** David C. Colston.

**Funding acquisition:** Vivian F. Go, William C. Miller.

**Investigation:** David C. Colston, Adams L. Sibley, William C. Miller.

**Methodology:** David C. Colston, Vivian F. Go, Clare Barrington, William C. Miller.

**Project administration:** David C. Colston, Vivian F. Go, Adams L. Sibley, Hannah M. Piscalko, William C. Miller.

**Resources:** Vivian F. Go, William C. Miller.

**Software:** Vivian F. Go, William C. Miller.

**Supervision:** Vivian F. Go, William C. Miller.

**Validation:** David C. Colston.

**Visualization:** Vivian F. Go.

**Writing – original draft:** David C. Colston, Vivian F. Go, Clare Barrington.

**Writing – review & editing:** David C. Colston, Vivian F. Go, Clare Barrington, Adams L. Sibley, Hannah M. Piscalko, Laura Limarzi-Klyn, William C. Miller.

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
