## [Decision Letter · Decision Letter 0]

25 Aug 2025

Dear Dr. Colston,

Thank you for submitting your manuscript to PLOS ONE. After careful consideration, we feel that it has merit but does not fully meet PLOS ONE’s publication criteria as it currently stands. Therefore, we invite you to submit a revised version of the manuscript that addresses the points raised during the review process.

We look forward to receiving your revised manuscript.

Kind regards,

Joseph Gregory Rosen

Academic Editor

PLOS ONE

“This work was supported by National Institutes of Health/National Institute on Drug Abuse Grant Numbers:UG3DA044822/UH3DA044822/ UH3DA044822-04S1 (Grants awarded to WCM, VFG) https://www.drugabuse.gov/.”

Reviewers' comments:

Reviewer's Responses to Questions

**Comments to the Author**

1. Is the manuscript technically sound, and do the data support the conclusions?

Reviewer #1: Yes

Reviewer #2: Partly

2. Has the statistical analysis been performed appropriately and rigorously?

Reviewer #1: Yes

Reviewer #2: N/A

3. Have the authors made all data underlying the findings in their manuscript fully available?

Reviewer #1: No

Reviewer #2: No

4. Is the manuscript presented in an intelligible fashion and written in standard English?

Reviewer #1: Yes

Reviewer #2: Yes

Reviewer #1: The authors present a paper titled “It’s a last straw situation”; Overdose Response and Preferences for Seeking Support from Emergency Services during Overdose in Rural Ohio.

This is an important qualitative study that explores a crucial, underexamined area: the real-world dynamics and decision-making processes behind whether people who use drugs (PWUD) in rural Appalachia call 911 during overdose events. The manuscript is well-organized, thoughtfully written, and grounded in strong public health relevance. With some strengthening of methods transparency, clearer demographic framing, and more focused theme organization, this paper will be an important contribution.

Here are some comments:

1. Introduction: Emphasize the unique contribution of this paper to make it clearer upfront in the Introduction. What gap in knowledge does this study fill that others don’t? Why is this particular rural Ohio context novel?

2. Method: Regarding coding - how many coders were involved? Was intercoder reliability assessed? How were disagreements resolved?

Regarding results - the manuscript tells us N = 44, but gives very little demographic context (e.g., age, gender, race, legal status, housing status). This weakens interpretation of findings, especially since issues like stigma, criminal-legal involvement, and healthcare access are highly intersectional. Please provide a summary table of participant characteristics to help contextualize the quotes and themes.

3. Discussion: Some recommendations—e.g., “ensure law enforcement doesn’t arrest people at the scene”—while valid (and policy-oriented) could be hard to implement without more nuance. I suggest expanding on what practical steps could improve GSL awareness, reduce stigma among EMS/police, and build trust. Could peer responders or harm reduction outreach workers play a role?

4. Other minor comments: The title is compelling, but the phrase “last straw situation” is buried in the text and not highlighted much. Either elevate its significance in the narrative or consider a more thematically appropriate title.

Also, the data availability note says data access is restricted due to human subject protections. This is reasonable, but if possible, consider sharing a codebook or de-identified quote bank for transparency.

Reviewer #2: Please see attachment. Further more- having a minimum character requirement for this field is preposterous, Its taken me 10 minutes to figure out what the problem is in submitting this form. sjdhaksjdhjaksdhajks

**Do you want your identity to be public for this peer review?** For information about this choice, including consent withdrawal, please see our Privacy Policy

Reviewer #1: No

Reviewer #2: No

---

## [Author Response · Author response to Decision Letter 1]

4 Nov 2025

Thank you to reviewers. We have attached detailed responses to questions, recommendations.

---

## [Editor Report · Decision Letter 1]

26 Nov 2025

“It’s a last straw situation”; Overdose Response and Preferences for Seeking Support from Emergency Services during Overdose in Rural Ohio

PONE-D-25-08847R1

Dear Dr. Colston,

We’re pleased to inform you that your manuscript has been judged scientifically suitable for publication and will be formally accepted for publication once it meets all outstanding technical requirements.

Kind regards,

Joseph Gregory Rosen

Academic Editor

PLOS ONE
---

## [Editor Report · Acceptance letter]

PONE-D-25-08847R1

PLOS One

Dear Dr. Colston,

I'm pleased to inform you that your manuscript has been deemed suitable for publication in PLOS One. Congratulations! Your manuscript is now being handed over to our production team.

Kind regards,

on behalf of

Dr. Joseph Gregory Rosen

Academic Editor

PLOS One